# Monitoring Alcohol Consumption in Slovak Cities during the COVID-19 Lockdown by Wastewater-Based Epidemiology

**DOI:** 10.3390/ijerph20032176

**Published:** 2023-01-25

**Authors:** Paula Bimová, Alexandra Tulipánová, Igor Bodík, Miroslav Fehér, Martin Pavelka, Sara Castiglioni, Ettore Zuccato, Noelia Salgueiro-González, Nina Petrovičová, Ján Híveš, Viera Špalková, Tomáš Mackuľak

**Affiliations:** 1Department of Inorganic Technology, Faculty of Chemical and Food Technology, Slovak University of Technology, Radlinského 9, 812 37 Bratislava, Slovakia; 2Institute of Chemical and Environmental Engineering, Faculty of Chemical and Food Technology, Slovak University of Technology, Radlinského 9, 812 37 Bratislava, Slovakia; 3Ministry of Health of the Slovak Republic, Limbová 2, 837 52 Bratislava, Slovakia; 4London School of Hygiene and Tropical Medicine, Faculty of Epidemiology and Population Health, Keppel Street, London WC1E 7HT, UK; 5Istituto di Ricerche Farmacologiche Mario Negri IRCCS, Department of Environmental Health Sciences, Via Mario Negri 2, 201 56 Milan, Italy; 6Department of Zoology and Fisheries, Faculty of Agrobiology, Food and Natural Resources, Czech University of Life Sciences, Kamýcka 129, 165 00 Praha 6, Czech Republic

**Keywords:** urban wastewater, alcohol, liquid chromatography-tandem mass spectrometry, COVID-19, lockdown

## Abstract

The consumption of alcohol in a population is usually monitored through individual questionnaires, forensics, and toxicological data. However, consumption estimates have some biases, mainly due to the accumulation of alcohol stocks. This study’s objective was to assess alcohol consumption in Slovakia during the COVID-19 pandemic-related lockdown using wastewater-based epidemiology (WBE). Samples of municipal wastewater were collected from three Slovak cities during the lockdown and during a successive period with lifted restrictions in 2020. The study included about 14% of the Slovak population. The urinary alcohol biomarker, ethyl sulfate (EtS), was analyzed by liquid chromatography-tandem mass spectrometry (LC-MS/MS). EtS concentrations were used to estimate the per capita alcohol consumption in each city. The average alcohol consumption in the selected cities in 2020 ranged between 2.1 and 327 L/day/1000 inhabitants and increased during days with weaker restrictions. WBE can provide timely information on alcohol consumption at the community level, complementing epidemiology-based monitoring techniques (e.g., population surveys and sales statistics).

## 1. Introduction

One of the most often used legal drugs in the world that can lead to addiction is alcohol [1]. The average per capita alcohol consumption varies worldwide, being the highest in the European Union with an annual consumption of about 11 L of pure ethanol per capita, followed by the United States with an annual consumption of 9.97 L of pure alcohol. The predominant religions in the Middle East tend to view alcohol consumption in a negative light which is evident in the regions with significantly low consumption levels (e.g., Afghanistan with 0.013 L of pure ethanol per capita). Average per capita alcohol consumption in the European Union decreased from 11.5 to 11.3 L of pure alcohol per capita between 2010 and 2016. The same source states that Slovakia has a somewhat average consumption, averaging 11.1 L of pure alcohol. [2]. At the beginning of the COVID-19 pandemic in 2020, governments around the world imposed state lockdowns and selfisolation measures on their citizens as a mechanism to challenge the spread of the disease. In Slovakia, the government implemented the closure of all nonessential services on 12 March 2020, including all nightclubs, restaurants, and pubs; consequently, the purchase of alcoholic beverages was limited to standard grocery stores. In this context, changes in the patterns of behavior associated with alcohol sales and consumption were expected and monitored [3]. Slovakia recorded the first case of COVID-19 on 6 March 2020. The Slovak government’s response was very fast, and the first set of control measures was already introduced on 12 March 2020. This included the closure of schools, universities, and nonessential shops and services. Churches were also closed, and mass events and gatherings were banned. People were requested not to meet with other households. Restaurant activity was restricted to only takeaway and delivery services. People were, however, allowed to leave their homes to go to work and for the purpose of recreational walking. A national stay-at-home order with no exceptions was briefly in place between 8 and 14 April 2020. Control measures were phased out in stages. The first stage (22 April) saw the opening of small and mid-sized nonessential shops (up to 300 sqm), albeit with a restricted capacity, and outdoor markets. The second stage (6 May) saw the opening of services, gastronomy, and churches and allowing wedding receptions, again with restricted capacity. The third stage (20 May) saw the opening of fitness, wellness, and large-size nonessential shops. Mass events (up to 1000 people) were allowed from 1 July. Furthermore, restrictions were imposed on arrivals into Slovakia. Returning citizens were asked to stay for 14 days in government-designated quarantine centers and take a mandatory swab test for COVID-19. A national test and trace program was quickly set up, and within the first weeks of the epidemic, Slovakia was able to test up to 10,000 people every day (Ministry of Health of the Slovak Republic).

Based on general population surveys and sales data, common techniques of alcohol use screening were used [4]. However, since these are dependent on customer statements and sales data do not always reflect immediate alcohol usage, they are intermittently not completely dependable [5]. A validated approach called wastewater-based epidemiology (WBE) can be used to estimate the real alcohol consumption by a population, complementing the common indicators and helping in the evaluation of alcohol policies [6]. WBE is based on the chemical analysis of specific human metabolites (biomarkers) in raw wastewater (WW), which should meet particular criteria, including being measured and present in WW, excreted in considerable amounts, and specific to human metabolism [7,8]. The current WBE method uses ethyl sulfate (EtS) as a reliable biomarker of alcohol consumption, as demonstrated in previous studies [5,9]. 

WBE has been successfully applied to evaluate alcohol consumption at national [5,6,10,11,12,13] and international levels [9] but also to assess changes in alcohol habits during the COVID-19 pandemic in Austria [14], Belgium [15], and Australia [16,17,18].

The aim of the presented study was to assess alcohol consumption in Slovakia during the COVID-19 pandemic-related lockdown using wastewater-based epidemiology (WBE).

As far as we know, no previous information about this topic can be found in the literature. It is important to apply the WBE to investigate the average alcohol consumption during this specific period to supply the other methods for the screening of alcohol consumption. The investigation covered three Slovak cities monitored in 2020 during the lockdown (April–May 2020) and during a successive period with milder restrictions (October–November 2020). These three cities were selected as a representative parts of the country. Bratislava and Košice are the two biggest cities in Slovakia, and Piešťany is a famous spa city, which can be considered as a good representative sample for a comparison of alcohol consumption between lockdown and normal days. WBE is a highly helpful instrument for delivering accurate and current information on human behaviors in a defined location, as shown by past research. In order to better focus on a national (or Slovak) preventive program, this study may offer further information on the epidemiological indicators that are already in use and also help with the application of preventive actions, e.g., in schools.

## 2. Participants and Methods

### 2.1. Participants

#### 2.1.1. Sampling

Using an automated sampler, raw WW samples were proportionally obtained from the inlets of 10 Slovak wastewater treatment plants (WWTPs) located in the cities of Bratislava, Košice, and Piešťany. A total of 40 samples were collected during the 2020 sampling program. Table 1 displays the collection dates. Aliquots of WW (50 mL) were taken every 15 min for 24 h, starting at 7:00, to obtain a daily representative sample (4.8 L). We took 0.5 L of 24 h composite samples that were frozen at −20 °C within 2 h of collection [19]. The sampling procedure was in accordance with the protocol established in previous studies in order to keep sampling uncertainty to a minimum [20,21,22]. Sampling personnel wore standard personal protective equipment (PPE) for wastewater sampling, such as long trousers, steel-capped boots, hard hats, safety glasses, and gloves to minimize potential exposure to infectious SARS-CoV-2.

#### 2.1.2. Characterization of the Analyzed Cities

Three cities in Slovakia were chosen based on population density (Table 1). Their combined population, which makes up about 14% of the nation’s total, is around 730,000 people. The population is a technical term used by WWTP staff to describe how many people are connected to the WWTP. The census data, or a figure calculated from hydrochemical parameters, can be used (e.g., biological oxygen demand—BOD or ammoniacal nitrogen). The population sizes employed in this investigation had already been gathered in earlier studies [12,23] using known techniques.

### 2.2. Methods

#### 2.2.1. Chemicals and Materials

Analytical standards of EtS sodium salt and Ets-d5 sodium salt (internal standard, IS) were acquired from Cerriliant (Round Rock, TX, USA) as solutions of 1 mg/mL (as ethyl sulfate) in methanol (MeOH). Before each analytical run, working solutions (ranging from 0.01 to 1 g/mL) were newly made with MeOH and kept at −20 °C in the dark. MeOH for pesticide analysis, ACN for LC-MS, and acetic acid (98%) were obtained from Fluka (Buchs, Switzerland) and from Carlo Erba (Cornaredo, Italy). Ultrapure water (Milli-Q) was directly obtained using a MILLI-RO PLUS 90 apparatus (Millipore, Molsheim, France).

#### 2.2.2. Sample Pretreatment and Instrumental Analysis

EtS was analyzed by liquid chromatography-tandem mass spectrometry (LC-MS/MS) using direct injection, according to a previously validated/published method [6,10,11]. Raw wastewater samples were filtered under vacuum using firstly 1.6 μm GF/A glass microfiber filters (Whatman, Kent, UK) and then 0.45 μm nitrocellulose filters (Millipore, Bedford, MA, USA). A small volume of filtered wastewater sample (0.5 mL) was centrifuged at 2500 rpm for 5 min; an aliquot of the supernatant (190 µL) was spiked with the internal standard EtS-d5 (spiked amount 10 ng) and transferred into a glass vial for LC-MS/MS analysis. LC-MS/MS analyses were carried out using an Agilent 1200 series LC system coupled to a triple quadrupole mass spectrometer (AB—Sciex API 5500) equipped with an electrospray ionization source, operating in the negative mode. LC separation was carried out with an Atlantis T3 2.1 mm × 150 mm, 3 µm column (Waters Corporation, Milford, MA, USA) using 0.1% acetic acid in Milli-Q water as mobile phase A and acetonitrile as phase B. The flow rate was 0.180 mL/min, and the injection volume was 4 μL. Quantitative analyses were performed in multiple reaction monitoring (MRM) mode, selecting two MRM transitions for EtS (125.1 > 97.1 and 125.1 > 80.1) and one for EtS-d5 (130.1 > 98.1). Retention time and relative ion abundances (MRM ratio) were used for the identification in wastewater samples, according to the European guidelines [24]. Data were acquired and processed using Analyst^®^ 1.6.1 software (AB Sciex). Quantification was carried out using the isotopic dilution method with six-point calibration curves (0–160 ng/mL; IS concentration: 50 ng/mL) prepared freshly before each analytical run. Procedural blanks (mineral water) were included in each batch to check for eventual contamination. 

#### 2.2.3. Estimation of Alcohol Use

The daily mass loads (g/day) of alcohol consumption were calculated by multiplying the measured EtS concentrations by the daily flow rate of each WWTP. An established technique that takes into account a correction factor (CF) of 3047 was used to back-calculate alcohol consumption [25]. To compare the results from the various cities, the volume of pure alcohol drank daily per capita (L/day) was calculated using alcohol density (0.789 kg/L). This figure was then adjusted to the population (L/1000 inh/day).

## 3. Results and Discussion

All samples were taken in 2020 during the lockdown days (April to May 2020), and the period with less restrictive measures included EtS (30 October–2 November), as reported in Table 2 and Table 3. The average daily consumption during lockdown ranged from 2.1 to 23.6 L/1000 inh/day in Košice, from 13.7 to 19.8 L/1000 inh/day in Bratislava Petržalka, from 8 to 15.2 L/1000 inh/day in Bratislava Central, and from 12.9 to 23.3 L/1000 inh/day in Piešťany. The average daily consumption during days with milder restrictions ranged from 20.4 to 28.9 L/1000 inh/day in Košice, from 16.1 to 327.3 L/1000 inh/day in Bratislava Petržalka, from 21.4 to 57.6 L/1000 inhabitants/day in Bratislava Central, and from 8.5 to 40.5 L/1000 inh/day in Piešťany. 

The highest consumption was found in Petržalka at the beginning of November (weekend 1–2 November 2020), which is more than three times higher than the results from a previous study (alcohol consumption in Bratislava during November 2017—average consumption was 49 L/1000 inh/day); this may have been caused due to a colder season and some specific events during that time also described in other studies. Specific events, such as Halloween or Christmas parties, which are typical in the winter season, occurred during the time when the restrictions were lighter [10,11,26]. 

Figure 1 depicts the average alcohol consumption in the three cities of Slovakia during both the lockdown and the time of more relaxed restrictions. In all cities, there was a striking reduction (15–80%) in alcohol consumption during lockdown restrictions.

The highest difference in alcohol consumption between lockdown and the days with milder restrictions can be seen in Bratislava–Petržalka (almost 80% decrease); the lowest difference was seen in Piešťany (15%). Piešťany is a spa city, which was also visited during the lockdown due to medical reasons, and this could be a reason for the lowest difference. On the other side, the decrease in alcohol during the lockdown in Peržalka might be the result of changing demographics with a lot of university students or people who were working from home.

Government interventions during the lockdown affected alcohol consumption mainly due to closed restaurants, pubs, and banned mass events. Restrictions on social interactions with nonfamily members and fewer opportunities for the social drinking of alcoholic beverages have reduced overall alcohol consumption. When compared to our previous results, alcohol consumption during the years 2017–2018 in several Slovak cities (e.g., Bratislava in 2018—30 L/1000 inh/day) was much higher than during the period with lockdown restrictions in 2020 (overall mean 11.9 L/1000 inh/day) [11].

Our results are in good agreement with a similar study performed in Innsbruck (Austria), where alcohol consumption was generally reduced during the lockdown days compared to the days without restrictions in 2019 [14]. The results obtained are also in good agreement with the study carried out in Adelaide (Australia) between April 2016–April 2020, which included the lockdown period. This study was realized in a catchment area of 1.1 million inhabitants. The results showed that alcohol consumption in April 2020 was the lowest since the beginning of the monitoring period (April 2016). The overall weekly consumption during lockdown decreased, but also the weekend increase was flatter than usual [18]. Another study provided in Australia also showed that during the implementation of the lockdown restrictions, consumption decreased, but right after reducing the restrictions, consumption returned to the levels before the pandemic or, in some instances, consumption became even higher [17]. The study provided in Belgium also showed a decrease in alcohol consumption right after the implementation of the COVID-19 restrictions [15].

## 4. Conclusions

In this study, WBE was used to investigate the pattern of alcohol consumption during the lockdown caused by the COVID-19 pandemic in selected Slovak cities. Temporal trends were identified by determining the alcohol biomarker EtS in wastewater. The obtained results show that alcohol consumption in selected cities during government restrictions decreased compared to the period of milder restrictions. Due to the shutdown of restaurant services and the prohibition of all kinds of events and social interactions with nonfamily members, a ca. 20% reduction in alcohol consumption was observed, and this seems reasonable. Despite the limitations of monitoring alcohol consumption by wastewater analysis (e.g., changing population, scant metabolism data, and uncertainties related to sampling and analytical methods), the results clearly show that WBE can be used to identify quarantine changes in lifestyle, behavior, and health that are associated with the consumption of illicit drugs or alcohol. 

## Figures and Tables

**Figure 1 ijerph-20-02176-f001:**
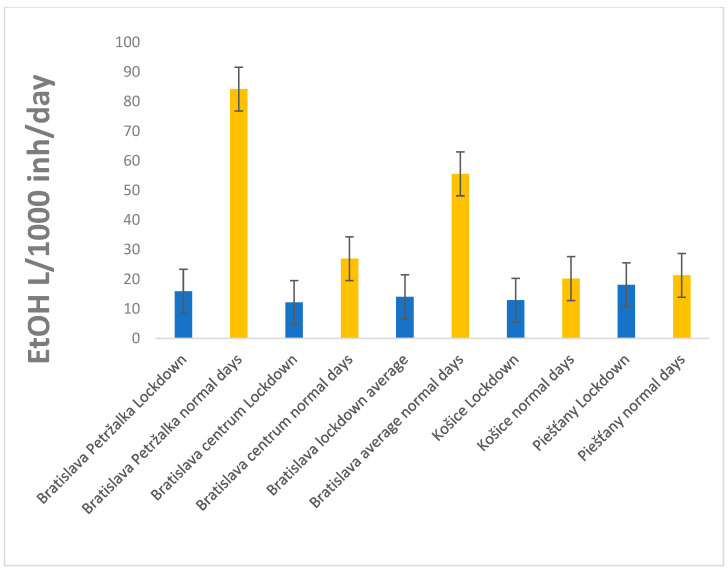
Average alcohol consumption (L/1000 inh/day) in three Slovak cities during the lockdown and a period with lighter restrictions in 2020 (marked as “normal” and orange color). Error bars represent the standard deviation of the values for each period.

**Table 1 ijerph-20-02176-t001:** Characteristics of selected WWTPs and sampling period.

WWTP	Population Connected (Inhabitants—Census)	Mean Daily Inflow [m^3^/day]	Sampling Period
Bratislava Central	450,000	124,000	**30/4/2020** **19/5/2020** **21/5/2020** 30–31/10/2020 1–2/11/2020
Bratislava Petržalka	125,000	32,500	**5/5/2020** **18/5/2020** **21/5/2020** 30–31/10/2020 1–2/11/2020
Košice	215,000	60,500	**21/4/2020** **20/5/2020** 31/10/2020 1/11/2020 18/11/2020 25/11/2020
Piešťany	30,000	15,000	**30/3/2020** **22/4/2020** 31/10.2020 2–3/11/2020

* Lockdown days marked in bold.

**Table 2 ijerph-20-02176-t002:** Estimated daily consumption of alcohol in three Slovak cities during normal days in 2020.

	Sampling Period	EtS [ng/mL]	Alcohol Consumption [L/1000 inh/day]	Alcohol Consumption [L/inh/day]
**Bratislava** **Central** **Bratislava** **Petržalka** **Košice** **Piešťany** **Overal mean (normal days)**	30/10/2020 31/10/2020 1/11/2020 2/11/2020 30/10/2020 31/10/2020 1/11/2020 2/11/2020 31/10/2020 1/11/2020 18/11/2020 25/11/2020 31/10/2020 2/11/2020 3/11/2020	49.7 21.3 41.4 18.4 16.4 20.9 64.5 141 20.6 16.9 26.6 16.8 4.84 22.0 13.5 32.1	57.6 24.7 48.4 21.4 16.1 48.5 149.7 327.3 25.2 20.8 28.9 20.4 8.5 40.5 21.4 30.4	0.058 0.025 0.048 0.021 0.016 0.049 0.150 0.327 0.025 0.021 0.029 0.020 0.009 0.041 0.021 0.030

**Table 3 ijerph-20-02176-t003:** Estimated daily consumption of alcohol in three Slovak cities during lockdown in 2020.

	Sampling Period	EtS [ng/mL]	Alcohol Consumption [L/1000 inh/day]	Alcohol Consumption [L/inh/day]
**Bratislava** **Central** **Bratislava** **Petržalka** **Košice** **Piešťany** **Overal mean (lockdown)**	30/4/2020 19/5/2020 21/5/2020 5/5/2020 18/5/2020 21/5/2020 21/4/2020 20/5/2020 30/3/2020 22/4/2020	21.2 14.7 8.82 12.1 11.7 19.0 2.22 23.9 18.8 12.6 14.5	15.2 13.3 8.0 14.2 13.7 19.8 2.1 23.6 23.3 12.9 11.9	0.015 0.013 0.008 0.014 0.014 0.020 0.002 0.024 0.023 0.013 0.012

## Data Availability

The data presented in this study are available within this article.

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
