# Peer review of "Monitoring Alcohol Consumption in Slovak Cities during the COVID-19 Lockdown by Wastewater-Based Epidemiology"

_ijerph, 2023, doi:10.3390/ijerph20032176_

Round 1

Reviewer 1 Report

Article Comments

Monitoring alcohol consumption in Slovak cities during the COVID-19 lockdown by wastewater-based epidemiology.

Introduction

1.- It is necessary to complement the current situation of alcohol consumption worldwide to give a more complete picture of the object of study.

2.- It is necessary to specify the objective of the investigation.

3.- It is necessary to strengthen the importance of research.

Materials and methods

4.- In this section, it is required to indicate the statistical treatment that was applied to the results of the physicochemical characterization.

5.- With the results obtained, can statistical comparisons be made to determine if there are significant differences in the per capita consumption of alcohol in the three cities studied, establishing a level of significance?

6.- It is not clear what the population served by each treatment plant is with respect to the three cities.

7.- What were the criteria for selecting these three cities?

8.- Check if the daily mass loads were obtained by multiplying or dividing the EtS concentration by the daily flow.

Results and Discussion

9.- It is recommended to strengthen the discussion of results with other investigations.

10.- It is necessary to expand the information cited in lines 173 to 177.

11.- In reference to the differences of 80% and 15% mentioned in lines 181 to 183, what are the reasons that originate this difference?

Reviewer 2 Report

This is a valued contribution to overall WBE. It supports its strength as a respected tool  in  alcohol consumption monitoring. However highly appreciated, this paper has many flaws.

1.     Bizova et al. submitted an awkward manuscript, at the very best. Quantity of COVID-19 related manuscripts increases rapidly (this is sometimes called "an epidemic of manuscripts)  and first I suspected that Journal and long turnaround time might be responsible for outdated references, but this paper was submitted 2 or 3 weeks ago. So, "negligence" of the Journal could nevertheless be the reason for not including Boogaerts et a.  from 2022 or Bade et al. from 2021. These references are essential in the context of lns 64/65.

2.     Asides, the lead-off paragraph on ethanol and its consumption is quite substandard in its present form. It is a bit inappropriate to say that "... alcohol is also known as ..."

3.     Using "WBE" instead of the term in full throughout the MS Is an issue of consistency.

4.     Subdivision in paragraph “materials and methods” should be divided as follows:

2. Participants and methods

2.1. Participants

            2.1.1. Sampling

            2.1.2. Characterization of the analyzed cities

2.2. Methods

            2.2.1. Chemicals

You have two sections named “sample…”, basically, they can be merged into one.

5.     “2.4. Lockdown restrictions in Slovakia” should be part of the introduction, around ln. 51. There is no need to stress it out as a separate subsection.

6.     In “fig. 1” differences between lockdown and a period with smoother restrictions in 2020 would be apparent more clearly if orange and blue columns were distributed otherwise (next one to another), and labeled less ambiguously.

7.     Limitations of this study???????????

Round 2

Reviewer 2 Report

I believe this is ok now.